# TESLA: TASK-WISE EARLY STOPPING AND LOSS AGGREGATION FOR DYNAMIC NEURAL NETWORK INFERENCE

## ABSTRACT

For inference operations in deep neural networks on end devices, it is desirable to deploy a single pre-trained neural network model, which can dynamically scale across a computation range without comprising accuracy. To achieve this goal, Incomplete Dot Product (IDP) has been proposed to use only a subset of terms in dot products during forward propagation. However, there are some limitations, including noticeable performance degradation in operating regions with low computational costs, and essential performance limitations since IDP uses hand-crafted profile coefficients. In this paper, we extend IDP by proposing new training algorithms involving a single profile, which may be trainable or pre-determined, to significantly improve the overall performance, especially in operating regions with low computational costs. Specifically, we propose the Task-wise Early Stopping and Loss Aggregation (TESLA) algorithm, which is showed in our 3-layer multilayer perceptron on MNIST that outperforms the original IDP by 32% when only 10% of dot products terms are used and achieves 94.7% accuracy on average. By introducing trainable profile coefficients, TESLA further improves the accuracy to 95.5% without specifying coefficients in advance. Besides, TESLA is applied to the VGG-16 model, which achieves 80% accuracy using only 20% of dot product terms on CIFAR-10 and also keeps 60% accuracy using only 30% of dot product terms on CIFAR-100, but the original IDP performs like a random guess in these two datasets at such low computation costs. Finally, we visualize the learned representations at different dot product percentages by class activation map and show that, by applying TESLA, the learned representations can adapt over a wide range of operation regions.

## 1    INTRODUCTION

Inference operations in deep neural networks on end devices, such as mobile phones, embedded sensors, IoT devices, etc., have recently received increasing attention including McMahan et al. (2016), Howard et al. (2017), and Teerapittayanon et al. (2017). In such applications, it is desirable to deploy a single pre-trained CNN model on end devices, while allowing multiple operating regions to meet different power consumption, latency, and accuracy requirements. To achieve this goal, McDanel et al. (2017a) proposed the incomplete dot product (IDP) operation, where only a subset of terms is used in dot products of forward propagation. From now on, $x$% dot product (DP), where $0 \leq x \leq 100$, means the $x$% of terms used in dot products. As illustrated in Figure 1, 50% DP means half of filters are used during forward propagation, and thus only half of the output channels are retained. To reduce the deviation induced by IDP, filters are prioritized from most important to the least important by pre-determined monotonically non-increasing profile coefficients (say, $\gamma_1, ..., \gamma_N$) during training. Therefore, IDP can be applied at inference time with dynamically-adjusted degrees of completeness (specified by the percentage of terms being used) to trade off accuracy slightly for lowered power consumption and reduced latency. Specifically, VGG-16 model with 50% DP achieves 70% in accuracy on the CIFAR-10 dataset compared to the standard network achieves only 35% accuracy when using the reduced channel set.

While the original IDP design seems promising, there are two limitations. First, since the training process aims at optimizing the loss function computed using all weights of the model (100% DP),

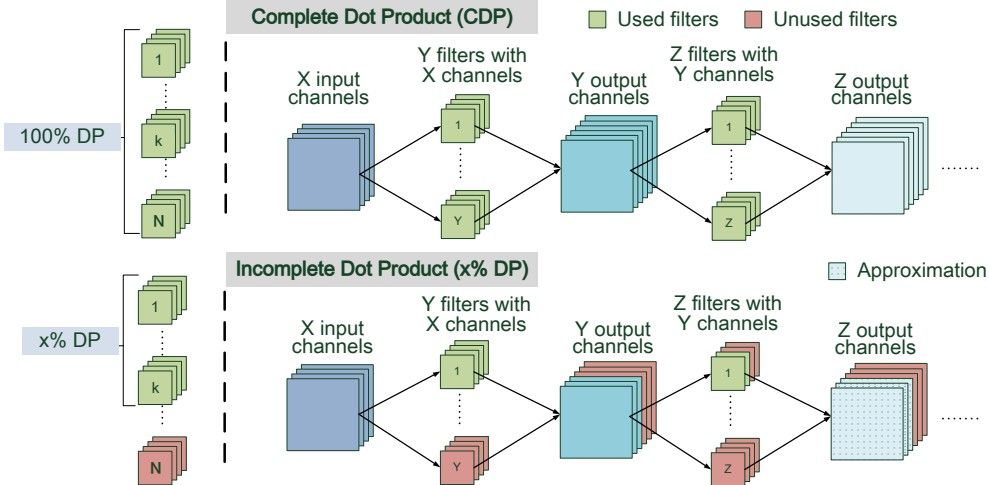

**Figure 1:** Comparison between complete dot product (CDP) and incomplete dot product (IDP) where x% DP implies only x% of filters are used to compute the corresponding output channel. Since only x% filters are unused, the resulting output is an approximation of the output under CDP.

there will be a mismatch between training and testing. It is no surprise that inference performance significantly decreases in low DP percentages and thus narrow the dynamic computation range. To mitigate this problem, the original IDP design utilizes the multiple-profile training strategy, where different profiles can be specified to focus on different dot product ranges. In such a multiple-profile training process, however, certain subset of weights will be frozen in each training stage corresponding to the profile being focused, and hence the overall performance may not be fully optimized. Besides, each profile needs to maintain a separate first and last layer for adjusting to its own dot product range, resulting in additional memory overhead. The second limitation relates to the pre-determined nature of profile coefficients. While there are multiple ways to set the profile based on different dynamic range requirements, the original IDP design did not focus on finding a single "best" profile that leads to the best performance. Instead, they use multiple hand-crafted profile coefficients, which make the system design less general among different applications, and hence may limit the overall performance of the system.

To reduce the mismatch between training and testing performances, we propose the Task-wise Early Stopping and Loss Aggregation (TESLA) algorithm, in which multiple loss functions are computed in different DP percentages. By gradually aggregating these loss functions in decreasing order of DP percentages as the objective function to be optimized, TESLA significantly improves testing performances in low DP percentages without compromising accuracy in medium to high DP percentages. The loss functions can also be aggregated in random order of DP percentages to make a variant of TESLA, called Randomized TESLA (R-TESLA), which enables better performances under pre-specified operating regions of end devices. Moreover, we relax the constraint of pre-determined profile coefficients and propose the alternate training procedure (ATP) to alternately train the profile coefficients along with weights of the model. By introducing trainable profile coefficients, customization among different applications can be achieved in a more generalized way, and the overall performance can also be further improved. This paper has made two major contributions: (1) We propose the Task-wise Early Stopping and Loss Aggregation (TESLA) algorithm and Randomized TESLA that can achieve dynamic scaling over a computation range in neural network inference without compromising accuracy. (2) We also propose the Alternate Training Procedure (ATP) that can learn the profile coefficients and the model weights simultaneously without the need of manual configuration of the profile coefficients.

## 2 INCOMPLETE NEURAL NETWORKS

Incomplete dot product (IDP) is a novel mechanism proposed by McDanel et al. (2017a) that can be applied to a hidden layer of MLPs or deep CNN models to dynamically lower the inference costs

by computing only a subset of terms in dot products during forward propagation. By introducing a set of non-increasing coefficients $\gamma_i$, referred to as a profile, to the channels during training, the channels will be ordered implicitly in non-increasing order from the most important to the least important. By simply dropping out less important channels at inference time, it suffices to train and deploy a single network, while still supporting different levels of computation scaling without compromising accuracy significantly. In this section, we briefly introduce the main concepts of IDP.

## 2.1 INCOMPLETE DOT PRODUCT OPERATION

Mathematically, for an IDP fully-connected layer with input dimension $N$ and output dimension $M$, the $j$-th output component $y_j$ is computed as

$$y_j = \sum_{i=1}^{N} \gamma_i w_{ji} x_i, \tag{1}$$

for $j \in \{1, 2, ..., M\}$, where $x_i$ is the $i$-th input component, $w_{ji}$ is the weight corresponding to the $j$-th output component and the $i$-th input component, and $\gamma_i$ is the $i$-th profile coefficient.

Similar expression can be derived for the IDP operation applied to a convolutional layer of CNN, as illustrated in Figure 1. For an IDP convolutional layer with number of input channels $N$ and number of output channels $M$, the $j$-th output channel $\mathbf{y}_j$ is computed as

$$\mathbf{y}_j = \gamma_j \sum_{i=1}^{N} \mathbf{f}_{ji} * \mathbf{x}_i, \tag{2}$$

for $j \in \{1, 2, ..., M\}$, where $\mathbf{f}_{ji} * \mathbf{x}_i$ denotes the convolution operation of the $i$-th input channel $\mathbf{x}_i$ and the $i$-th channel of the $j$-th filter $\mathbf{f}_{ji}$, and $\gamma_j$ is the profile coefficient for the $j$-th filter. Note that, instead of applying profile coefficients depthwise on each filter before convolution as is the case in the original IDP design, we multiply each $\gamma_j$ to each output channel after a complete convolution to produce $\mathbf{y}_j$. These two approaches, however, are equivalent with negligible difference induced by the first hidden layer. Since the output channels $\mathbf{y}_j$'s become input channels $\mathbf{x}_i$'s to the next layer, applying $\gamma_j$'s to $\mathbf{y}_j$'s is equivalent to applying them into the convolution operation in the next layer.

To compute IDP with a target dot product percentage, a truncated version of Eq. 1 or Eq. 2 replaces the original computation to keep only a subset of the beginning terms. As for the case with all terms are kept, we refer to such operations as complete dot product (CDP) or 100% DP, interchangeably. Note that in the training process in the original IDP design, only CDP is used.

## 2.2 MULTIPLE-PROFILE INCOMPLETE NEURAL NETWORKS

In the work of McDanel et al. (2017a), several profile coefficients are proposed and applied in a pre-determined manner. When only a single profile is applied to the model, the trade-off between computation range and performance in high DP percentage regions is also demonstrated. Generally, the faster the profile coefficients decrease, the larger computation range can be achieved, at the expense of a performance degradation in high DP percentage regions. To cover a larger computation range while maintaining the performance in high DP percentage regions, McDanel et al. (2017a) further introduced the multiple-profile incomplete neural networks (MP-IDP), where different profiles can be specified to focus on different DP ranges. During training, all the specified profiles are applied in increasing order of their operating DP ranges. When a profile is applied, only weights corresponding to its operating DP range will be updated, leaving weights corresponding to lower DP percentages frozen since they have been trained in previous stages, and weights corresponding to higher IDP percentages set to zeros since they will be trained in later stages. In such a stage-by-stage training process, the overall performance may not be fully optimized.

## 3 TASK-WISE EARLY STOPPING AND LOSS AGGREGATION

As discussed in Section 2, in the original IDP design, CDP is used during training but IDP is applied at inference time. This mismatch leads to a noticeable degradation in inference performance, especially in low DP percentages. To mitigate this problem, we propose the Task-wise Early Stopping

and Loss Aggregation (TESLA) algorithm. In this paper, a *task* is defined as the learning process that uses only a subset of weights determined by a DP percentage to learn the optimal representations. For example, a task of 50% DP implies that the first half of network weights are used for dot product computations and thus only these 50% of weights will be updated while conducting back-propagation. With TESLA, we can optimize a network by tasks with different DP percentages to support various levels of computation scaling and meanwhile reduce the mismatch between training and inference. The design of TESLA is described as follows.

### 3.1 TASK-WISE EARLY STOPPING

Since tasks with different DP percentages may have different learning difficulties and convergence rates, we apply an early stopping mechanism to automatically adjust the learning processes of tasks. Specifically, we keep all hyper-parameters unchanged except the numbers of epoches, which are controlled by the early stopping mechanism that halts the training process as long as the task performance has not been improved for a certain number of iterations. For example, considering two tasks, one using 70% DP (task 1) and the other using 40% DP (task 2), we first optimize task 1 and then switch to optimize task 2 until the optimization process of task 1 reaches the early stopping criterion. With this task-wise early stopping, we are able to optimize all the tasks sequentially, and each task initializes its model using the weights that have been optimized for all previous tasks. However, the weights used in task 2 is exactly a subset of weights used in task 1 such that the optimization process of task 2 may contaminate the well-trained weights for task 1. To reduce this unexpected disturbance while learning multiple tasks, some kinds of loss aggregation are needed to learn a new task without sacrificing the performance of all the past tasks too much.

---

**Algorithm 1** Task-wise Early Stopping and Loss Aggregation, TESLA

---

1: Input: a task set in decreasing order, $T = L_i$; aggregation coefficient $\alpha$
2: Initialization: $L_1^{obj} \leftarrow L_1$ and $i \leftarrow 1$
3: **while** $i \leq size(T)$ **do**
4:     optimize $L_i^{obj}$ until meeting early stopping criteria
5:     $L_{i+1}^{obj} \leftarrow \alpha \times L_{i+1} + (1 - \alpha) \times L_i^{obj}$
6:     $i \leftarrow i + 1$
7: **end while**

---

**Algorithm 2** Randomized TESLA, R-TESLA

---

1: Input: a task set in any order, $T = L_i$; allowable epoch, $max\_epoch$; aggregation coefficient $\alpha$
2: Initialization: $L_1^{obj} \leftarrow L_1$, $i \leftarrow 0$, and $n \leftarrow 0$
3: **while** $n \leq max\_epoch$ **do**
4:     optimize $L_i^{obj}$ until meeting early stopping criteria, which takes $n\_epochs$
5:     Sample a task called $L_k$
6:     $L_{i+1}^{obj} \leftarrow \alpha \times L_k + (1 - \alpha) \times L_i^{obj}$
7:     $i \leftarrow i + 1$
8:     $n \leftarrow n + n\_epcohs$
9: **end while**

---

### 3.2 TASK-WISE LOSS AGGREGATION

Task-wise loss aggregation is therefore proposed to jointly learn the shared representation for all tasks. By considering one new task at a time, we add the loss of the new task into the current objective function and optimize the aggregated objective function such that tasks are optimized *incrementally* and *jointly*. The aggregated objective function can be expressed as

$$L_1^{obj} = L_1 \ \text{ and } \ L_{i+1}^{obj} = \alpha \times L_{i+1} + (1 - \alpha) \times L_i^{obj} \ , \ \ \forall i = 1, \cdots, N - 1 \qquad (3)$$

where $\alpha$ is the aggregation coefficient shared by all subsequent tasks and greater $\alpha$ implies that we care more about the optimization of the new task. As a consequence, the objective function in the whole learning process is an affine combination of the losses of currently considered tasks. By

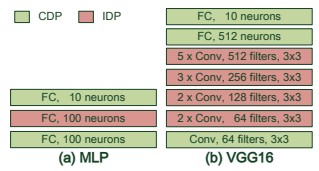

**Figure 2:** Network structures in study

**Table 1:** Hyper-parameters in Experiments

| Experiment | MLP on MNIST | VGG-16 on CIFAR-10 | VGG-16 on CIFAR-100 |
|---|---|---|---|
| Tasks at DP % | 100, 70, 40, 10 | 100, 50, 20 | 100, 70, 50, 30 |
| Learning rate | 0.001 | 0.004 | 0.004 |
| Optimizer | Adam | SGD momentum = 0.9 | SGD momentum = 0.9 |
| Batch size | 28 | 32 | 64 |
| Aggregation coefficient | 0.5 | 0.5 | 0.5 |
| TESLA stopping criteria | not improve in 4 epochs | not improve in 4 epochs | not improve in 4 epochs |
| R-TESLA stopping criteria | # epochs over 50 | # epochs over 35 | # epochs over 35 |
| Initial weights | random | pre-trained on ImageNet | pre-trained on ImageNet |

task-wise loss aggregation, these losses are aggregated incrementally and can be jointly optimized to learn a shared representation to be relevant to all tasks.

### 3.3 TESLA AND RANDOMIZED TESLA

**Task-wise Early Stopping and Loss Aggregation, TESLA.** We integrate task-wise early stopping and task-wise loss aggregation as TESLA to learn dynamic representations in neural networks. The entire training process optimizes all tasks in an arbitrary order. It is obvious that we have several options to order tasks in (i) increasing, (ii) decreasing, or (iii) random DP percentages. Recall that we add a non-increasing coefficients to prioritize terms in computing dot product, and thus the beginning terms, e.g. at 10% DP, are more important than the terms at last 10% terms. Therefore, discarding the terms from the end is less harmful to the optimized parameters, so TESLA is designed to optimize tasks in decreasing order of DP percentages. The TESLA algorithm is shown in Algorithm 1.

**Randomized Task-wise Early Stopping and Loss Aggregation, R-TESLA.** Here *Randomized* means that tasks are optimized in random order. The benefits of R-TESLA is two fold. First, R-TESLA provides an opportunity to turn attention back to optimize a task which had been halted before, and allows to finetune the weights, which may have been contaminated by other tasks. Second, unlike TESLA that optimizes each task only once, R-TESLA allows each task to be optimized for multiple times, which can be specified by a customized task distribution derived from the behavioral statistics of users or the specification of hardware design. The detailed procedures of R-TESLA are in Algorithm 2.

### 3.4 TRAINABLE PROFILE COEFFICIENTS

In this section we propose to learn profile coefficients along with weights of the model alternately. We initialize all coefficients as one and as long as any update of profile coefficients, we manually clip the coefficients to keep the non-increasing property. The alternate training procedure (ATP) relaxes the constraint of fixed coefficients and we demonstrate the feasibility of ATP in the experiment of the MLP model on MNIST dataset in Section 4.

## 4 EXPERIMENTS

In this section, we demonstrate the effectiveness of using TESLA and R-TESLA to learn dynamic representations in MLP and CNN models, with the widely-used datasets MNIST, CIFAR-10, and CIFAR-100. Figure 2 shows the network architectures in study. Note that while working on CIFAR-100, the last fully connected layer of Figure 2(b) is replaced by a single 100-class classifier. Here we compare TESLA and R-TESLA with the original IDP design proposed by McDanel et al. (2017a) over a range of dynamic scaling during inference. All hyper-parameters and experiment settings are summarized in Table 1.

### 4.1 MULTILAYER PERCEPTRONS

First, we consider a 3-layer MLP model, in which the IDP operation is applied to the first hidden layer, as shown in Figure 2(a), and evaluate on the MNIST dataset. In this experiment, we define four tasks that optimize the model at 10%, 40%, 70%, and 100% DP, respectively. It is noteworthy that defining too many tasks in our experiment would not benefit much, since there must be a large amount of shared parameters among tasks which makes the model vulnerable to overfitting.

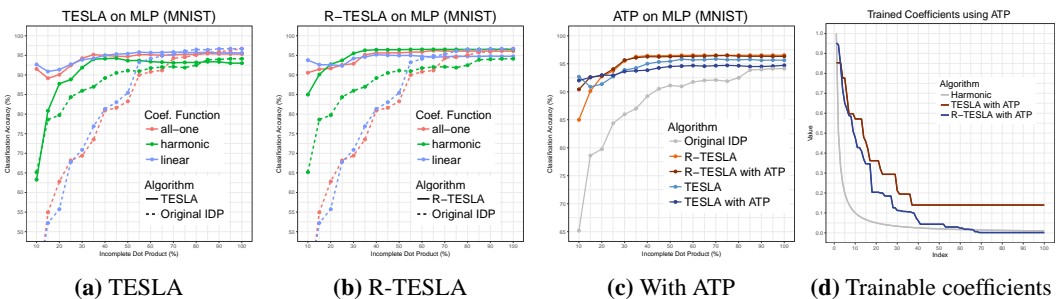

**(a)** TESLA  **(b)** R-TESLA  **(c)** With ATP  **(d)** Trainable coefficients

**Figure 3:** Performance comparisons by a MLP model over the MNIST dataset

**TESLA versus original IDP.** We compare TESLA and the original IDP design under various profiles. Figure 3(a) shows that at 20% DP, the original IDP achieves 80%, 63% and 55% accuracy for the harmonic, all-one, and linear profiles respectively but TESLA keeps at least 88% accuracy for all profiles at 20% DP and reaches average accuracy of 94.7% using the linear profile. Most importantly, compared to the original IDP, TESLA performs only about 1% worse in accuracy at 100% DP but gains a significant improvement from 50% to 90% in accuracy at 10% DP, which is an acceptable trade-off under practical applications.

**R-TESLA versus TESLA and original IDP.** Figure 3(b) shows that R-TESLA outperforms the original IDP by a large margin and R-TESLA has comparable performance with TESLA in most cases. R-TESLA with the harmonic profile leads to the best average accuracy of 95.2% in this experiment. By observing the optimization progress, we find that TESLA achieves its best result after completing the last task thanks to its ordinal optimization. On the other hand, we cannot ensure that R-TESLA can make the ultimate model retains the best dynamic representations due to its random nature.

**Learn profile coefficients by ATP.** Here we demonstrate the feasibility of learning profile coefficients along with weights. From Figure 3(c), with the help of trainable profile coefficients, both TESLA and R-TESLA further boost by 1% in average, and we also observe that the learned profile coefficients are similar to harmonic ones as shown in Figure 3(d). This may support why performance of harmonic coefficients is the best in the original IDP. By allowing coefficients to be trainable, it is no longer to require hand-crafted profile coefficients and determine the best profile coefficients by extensive experiments.

## 4.2 CONVOLUTIONAL NEURAL NETWORKS

We choose the known VGG-16 model pre-trained on ImageNet to evaluate over CIFAR-10 and CIFAR-100 dataset so that the last few dense layers are replaced by a 10-class classifier and a 100-class classifier respectively. Here we use the linear profile coefficients to compare: (i) the original IDP design, (ii) multiple-profile IDP design (MP-IDP) as proposed in McDanel et al. (2017a), (iii) TESLA, and (iv) R-TESLA. The experimental results are summarized below.

**VGG-16 on CIFAR-10.** According to Figure 4(a), the performance of original IDP by all-one coefficients drops much faster than that by linear coefficients. Appling all-one coefficients is equivalent to using the original VGG-16 network; however, linear profile coefficients implicitly encourages networks to learn channel importance in order, and also brings about that pruning away later channels at different DP percentages does not hurt the performance that much. With the use of multiple profiles, MP-IDP does enlarge the computational range with an increase in accuracy to 75% at 50% DP. Furthermore, the proposed algorithms, TESLA and R-TESLA, boost the accuracy to reach 85% at 50% DP, and an even higher accuracy at 100% DP.

Following the previous experiment, here we augment another new task of 20% DP and observe whether TESLA can leverage up the performance at low DP percentages by adding a task of a low DP percentage. Figure 4(b) shows that TESLA and R-TESLA greatly widens the computational ranges by making accuracy reaching 75% at 20% DP. We contribute this effect to applying TESLA and R-TESLA in decreasing order of dot product percentages so that the representation learned at 100% DP drives the training of representation at 50% DP, which also makes the representation

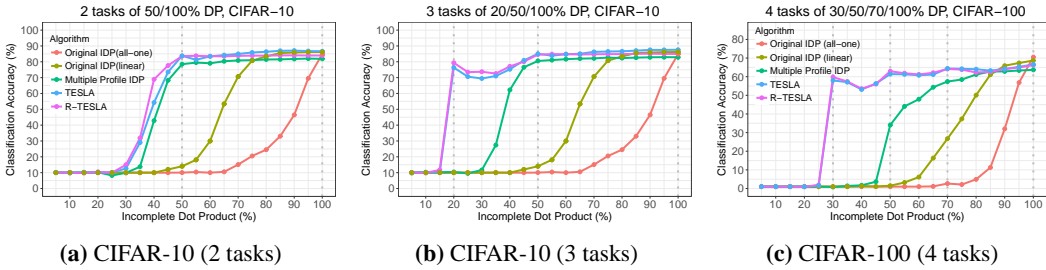

**Figure 4:** Performance comparisons by the VGG-16 model over the CIFAR-10 and CIFAR-100 dataset

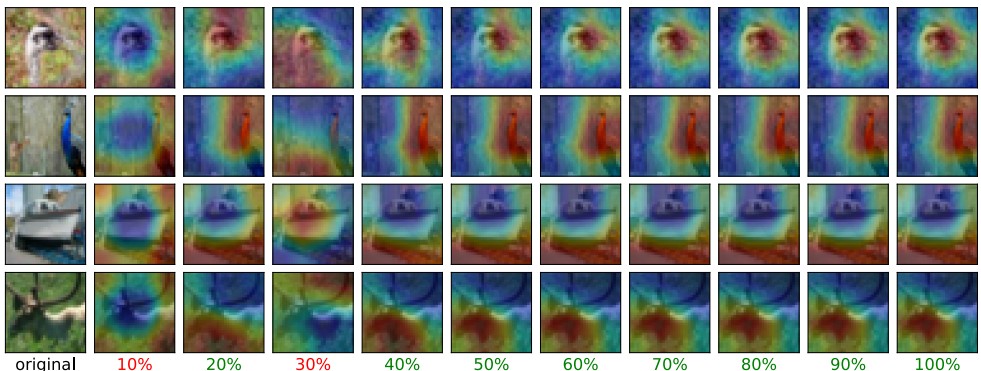

**Figure 5:** CAMs at different DP percentages. Red colored text means wrong prediction and green colored text means correct prediction.

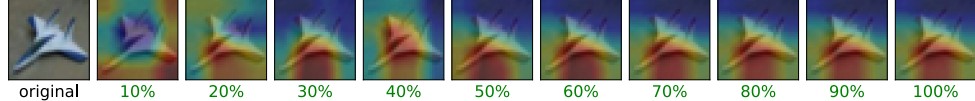

**Figure 6:** CAMs of a testing image that is correctly classified at all specified DP percentages.

much easier to be learned at 20% DP. Compared to TESLA and R-TESLA, MP-IDP trains models in increasing order of DP percentages and thus MP-IDP doesn't see much improvement at lower IDP percentages although adding another task at 20% DP.

**VGG-16 on CIFAR-100.** To sufficiently illustrate the effectiveness of the proposed approaches, we evaluate over a larger dataset, CIFAR-100. Figure 4(c) shows the performance of TESLA and R-TESLA still keeps around 60% accuracy from 30% to 50% DP, which outperforms either original IDP or MP-IDP by a significant margin, which is consistent with the result of CIFAR-10. Specifically, both TESLA and R-TESLA sacrifice about 4% accuracy at 100% DP but gain a great improvements of 60% accuracy in low DP percentages.

**CAM visualization.** We visualize what the model sees at different DP percentages by deploying the Class Activation Mapping (CAM) technique introduced by Zhou et al. (2016a). A resulting CAM indicates how much each location contribute to the final class prediction. In this stage, we replace the max-pooling layers with average-pooling layers and train the VGG-16 network with linear coefficients optimized at 20%, 50%, 100% DP. From CAMs at different DP percentages, we found that the network is easier to make wrong prediction at 10% and 30% DP but still makes correct prediction at 20% DP as shown in Figure 5 since the representations at 20% DP are optimized. This finding implies that we can specify any DP percentages to be optimized for satisfying custom requirements. Compared to Figure 6, we also notice that the CAMs at 10% DP are almost the same no matter the correctness of predictions, which indicates too limited capacity to capture meaningful patterns, and thus the network at 10% DP behaves like a random guess.

## 5 RELATED WORK

Our work is rooted from IDP proposed by McDanel et al. (2017a), which, in addition to MLPs and regular CNNs, can also be used in conjunction with other variants of convolutional layers, such as separable convolution layer Howard et al. (2017) and binary convolutional layer McDanel et al. (2017b). As discussed throughout this paper, our work extends the original IDP design by proposing new training algorithms involving a single profile, which may be trainable or pre-determined, to significantly improve the overall performance, especially in low DP percentages.

Network pruning is a widely-studied area that also aims at compressing the CNN models. Early works of network pruning construct a threshold for dropping weights by information obtained from Hessian matrix or inverse Hessian matrix in LeCun et al. (1990); Hassibi & Stork (1993), which adds memory and computation costs. In most of the recent works, magnitude-based pruning and recovering are incorporated to compensate the potential loss incurred by inadequate pruning. For example, Guo et al. (2016) introduces the splicing operation to enable connection recovery, and Han et al. (2016) directly makes the network dense again. Li et al. (2016) also prune filters in CNNs based on magnitude, but the number of filters pruned away in each layer is decided by layer-wise sensitivity. Besides magnitude-based pruning, a Taylor expansion-based criterion is introduced in Molchanov et al. (2016) to approximate the change in the cost function induced by pruning. In addition to network pruning, some works focus on low-rank decomposition for network compression. For example, Denton et al. (2014) and Jaderberg et al. (2014) approximate the weight matrix into low-rank components by minimizing the reconstruction error. Yu et al. (2017) further decomposes the weight matrix into its low-rank and sparse component. Other works focus on grouping similar weights, such as quantization by Han et al. (2015), Gong et al. (2014), and Zhou et al. (2017) and weight sharing by Ullrich et al. (2017), aiming at reducing the level of redundancy and the required storage. Yet another approach introduces group sparsity regularizer to constrain the structure of the model in Wen et al. (2016), Zhou et al. (2016b), and Alvarez & Salzmann (2016).

While all the above techniques are promising in reducing the size of the networks, none of them supports dynamic adjustment during inference as IDP does. Furthermore, most of the above techniques involve retraining the model iteratively, resulting in computational overhead. In our proposed work, the goal of efficient inference with dynamic adjustment can be readily fulfilled by training a single model at once, and the effectiveness is expected to be further improved by incorporating with other techniques listed above.

## 6 CONCLUSION

In this paper, we extend the idea of incomplete dot product (IDP) by proposing the Task-wise Early Stopping and Loss Aggregation (TESLA) algorithm to significantly improve the performance of neural networks with dynamically computation regions at inference time without significantly compromising accuracy. A task is defined as the learning process that uses only a subset of weights specified by a DP percentage to learn the optimal representations of the network. By introducing non-increasing profile coefficients to prioritize weights or filters during training, TESLA can be used to optimize multiple tasks in decreasing order of DP percentages by aggregating the their loss functions. Additionally, we propose Randomized TESLA (R-TESLA) which optimizes tasks in random order, and show that both TESLA and R-TESLA outperform original IDP and multiple-profile IDP significantly. The visualization of the class activation maps (CAMs) provide a strong evidence that the representations learned by TESLA allow dynamically scaling across a computation range to meet various power consumption, latency and accuracy requirements on end devices.

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
