# OpenReview forum: "TESLA: Task-wise Early Stopping and Loss Aggregation for Dynamic Neural Network Inference"
_ICLR.cc/2018/Conference — Reject_

### Official Review · AnonReviewer1 · 2017-11-27

**Rating:** 4
**Confidence:** 4

**Review:**

An approach to adjust inference speed, power consumption or latency by using incomplete dot products McDanel et al. (2017) is investigated.

The approach is based on `profile coefficients’ which are learned for every channel in a convolution layer, or for every column in the fully connected layer. Based on the magnitude of this profile coefficient, which determines the importance of this `filter,’ individual components in a neural net are switched on or off. McDanel et al. (2017) propose to train such an approach in a stage-by-stage manner.

Different from a recently proposed method by McDanel et al. (2017), the authors of this submission argue that the stage-by-stage training doesn’t fully utilize the deep net performance. To address this issue a `loss aggregation’ is proposed which jointly optimizes a deep net when multiple fractions of incomplete products are used.

The method is evaluated on the MNIST and CIFAR-10 datasets and shown to outperform work on incomplete dot products by McDanel et al. (2017) by 32% in the low resource regime.

Summary:
——
In summary, I think the paper proposes an interesting approach but more work is necessary to demonstrate the effectiveness of the discussed method. The results are preliminary and should be extended to CIFAR-100 and ImageNet to be convincing. In addition, the writing should be improved as it is often ambiguous. See below for details.

Review:
—————
1. Experiments are only provided on very small datasets. According to my opinion, this isn’t sufficient to illustrate the effectiveness of the proposed approach. As a reader I wouldn’t want to see results on CIFAR-100 and ImageNet using multiple network architectures, e.g., AlexNet and VGG16.

2. Usage of the incomplete dot product for the fully connected layer and the convolutional layer seems inconsistent. More specifically, while the profile coefficient is applied for every input element in Eq. (1), it’s applied based on output channels in Eq. (2). This seems inconsistent and a comment like `These two approaches, however, are equivalent with negligible difference induced by the first hidden layer’ is more confusing than clarifying.

3. The writing should be improved significantly and statements should be made more precise, e.g., `From now on, x% DP, where \leq x \geq 100, means the x% of terms used in dot products’. While sentences like those can be deciphered, they aren’t that appealing.

4. The loss functions in Eq. (3) should be made more precise. It remains unclear whether the profile coefficients and the weights are trained jointly, separately, incrementally etc.

5. Algorithm 1 and Algorithm 2 call functions that aren’t described/defined.

6. Baseline numbers for training on datasets without incomplete dot products should be provided.

---

> ### Author Response · Authors · 2018-01-05
> **Responses to the questions in review**
>
> Thanks for your comments. The followings are the response to your questions in the review:
>
> 1. Application to another dataset, VGG-16 on CIFAR-100
> In this revision, we evaluate the effectiveness of TESLA over a large dataset, CIFAR-100, and TESLA does outperform the original IDP design by a great margin, 60% accuracy, at low computation costs. Please find the experiment result on CIFAR-100 in Figure 4.
>
> 2. In actual, for both fully connected layer and convolution layer, the profile coefficient is applied equivalently. For example, in a fully connected layer, an input element first multiplies its weights and then multiplies its corresponding profile coefficients.
>
> 4. It is worthy to clarify that TESLA takes one new task into consideration at one time, and aggregate the loss of that new task into the current objective function and try to optimize the aggregated loss until meeting the early stopping criterion. Once early stopped, we add another new task and repeat the process until all tasks are optimized.
>
> 3, 5 and 6. Thanks for pointing out these errors. We have corrected them in this revision.
>
> Greatly thanks for your valuable comments.

---

### Official Review · AnonReviewer3 · 2017-11-29
**TESLA: Task-wise Early Stopping and Loss Aggregation for Dynamic Neural Network Inference**

**Rating:** 5
**Confidence:** 2

**Review:**

This paper presents a modification of a numeric solution: Incomplete Dot Product (IDP), that allows a trained network to be used under different hardware constraints. The IDP method works by incorporating a 'coefficient' to each layer (fully connected or convolution), which can be learned as the weights of the model are being optimized. These coefficients can be used to prune subsets of the nodes or filters, when hardware has limited computational capacity.

The original IDP method (cited in the paper) is based on iteratively training for higher hardware capacities. This paper improves upon the limitation of the original IDP by allowing the weights of the network be trained concurrently with these coefficients, and authors present a loss function that is linear combination of loss function under original or constrained network setting. They also present results for a 'harmonic' combination which was not explained in the paper at all.

Overall the paper has very good motivation and significance.
However the writing is not very clear and the paper is not self-contained at all. I was not able to understand the significance of early stopping and how this connects with loss aggregation, and how the learning process differs from the original IDP paper, if they also have a scheduled learning setting.

Additionally, there were several terms that were unexplained in this paper such as 'harmonic' method highlighted in Figure  3. As is, while results are promising, I can't fully assess that the paper has major contributions.

---

> ### Author Response · Authors · 2018-01-05
> **Difference between the original IDP paper**
>
> Thanks for your detailed comments.
>
> 1. To answer the question about "how the learning process differs from the original IDP paper, if they also have a scheduled learning setting.":
>
> (i) original IDP only trains the network using complete dot product, so, in our terminology, original IDP optimizes one task of 100% DP. And that is why original IDP does not perform well at inference time if low computation costs. On the other hand, TESLA adds a new task at a time and tries to optimize multiple tasks incrementally and jointly for the sake of improving the performance of the new task without contaminating the performance of the past tasks.
>
> (ii) A scheduled learning is similar but not equivalent to the effect of our early stopping design. For example, ideally, task A needs to 10 epochs to reach the optimal performance. If only allocates less than 10 epochs on task A, the model will underfit in task A; if allocates more than 10 epochs, says 20 epoches, the model will overfit. Thus, we believe the early stopping is a better way to adapt across tasks with different convergence rates.
>
> 2. Revision on the description of TESLA
> We have revised our paper and elaborate the details of the TESLA algorithm, which was originally cut for space considerations. Briefly, TESLA is designed to optimize multiple tasks incrementally and jointly by:
>
> (i) Task-wise early stopping: due to different learning difficulties and convergence rates of tasks, we apply an early stopping mechanism to adjust the training process of each task to reduce overfitting and optimize across all tasks sequentially.
>
> (ii) Task-wise loss aggregation: because of shared weights between any two tasks, when adding a new task at a time, we aggregate the loss of the new task into the current objective function and optimize jointly.
>
> Please check the revised section 3 to better understand the design intuition of our algorithm.
>
> 3. Application to another dataset, CIFAR-100
> In this revision, we also evaluate the effectiveness of TESLA over a large dataset, CIFAR-100, and TESLA does outperform the original IDP design. Please find the experiment result in Figure 4.
>
> Thank you.

---

### Official Review · AnonReviewer2 · 2017-12-01
**A potentially useful method, but not well motivated or explained**

**Rating:** 4
**Confidence:** 2

**Review:**

The authors propose a method for reducing the computational burden when performing inference in deep neural networks. The method is based a previously-developed approach called incomplete dot products, which works by pruning some of the inputs in the dot products via the introduction of pre-specified coefficients. The authors of this paper extend the method by introducing a task-wise learning procedure that sequentially optimizes a loss function for decreasing percentage of included features in the dot product.

Unfortunately, this paper was hard to follow for someone who does not actively work in this field, making it hard to judge if the contribution is significant or not. While the description of the problem itself is adequate, when it comes to describing the TESLA procedure and the alternative training procedure, the relevant passages are, in my opinion, too vague to allow other researchers to implement this procedure.

Positive points:
- The application seems relevant, and the task-wise procedure seems like an improvement over the original IDP proposal.
- Application to two well-known benchmarking datasets.

Negative points:
- The method is not described in sufficient detail to allow reproducibility, the algorithms are no more than sketches.
- It is not clear to me what the advantage of this approach is, as opposed to alternative ways of compressing the network (e.g. via group lasso regularization), or training an emulator on the full model for each task.

Minor point:
- Figure 1 is unclear and requires a better caption.

---

> ### Author Response · Authors · 2018-01-05
> **Thanks for your review and our paper is updated**
>
> Thanks for your comments.
>
> 1. Revision on the description of TESLA
> We have revised our paper and elaborate the details of the TESLA algorithm, which was originally cut for space considerations. Briefly, TESLA is designed to optimize multiple tasks incrementally and jointly by:
>
> (a) Task-wise early stopping: due to different learning difficulties and convergence rates of tasks, we apply an early stopping mechanism to adjust the training process of each task to reduce overfitting and optimize across all tasks sequentially.
>
> (b) Task-wise loss aggregation: because of shared weights between any two tasks, when adding a new task at a time, we aggregate the loss of the new task into the current objective function and optimize jointly.
>
> Please check the revised section 3 to better understand the design intuition of our algorithm.
>
> 2. Application to another dataset, CIFAR-100
> In this revision, we also evaluate the effectiveness of TESLA over a large dataset, CIFAR-100, and TESLA does outperform the original IDP design. Please find the experiment result in Figure 4.

---

### Decision · Program_Chairs · 2018-01-29
**ICLR 2018 Conference Acceptance Decision**

**Decision:**

Reject

**Comment:**

General consensus among reviewers that paper does not meet criteria for publication.

Pro:
- Improvement over the original IDP proposal.
- Some promising preliminary results.

Con:
- Insufficient comparison to other methods of network compression,
- Insufficient comparison to other datasets (such as ImageNet)
- Insufficient evaluation on variety of other models
- Writing could be more clear